# Imaging microglia surveillance during sleep-wake cycles in freely behaving mice

Xiaochun Gu[1,2,3,4]*[†], Zhong Zhao[5,6][†], Xueli Chen[6], Lifeng Zhang[6], Huaqiang Fang[6], Ting Zhao[1], Shenghong Ju[3], Weizheng Gao[7], Xiaoyu Qian[7], Xianhua Wang[6,8], Jue Zhang[7], Heping Cheng[6,8,9]*

[1]PKU-Nanjing Institute of Translational Medicine, Nanjing Raygen Health, Nanjing, China; [2]National Platform for Medical Engineering Education Integration, Southeast University, Nanjing, China; [3]Jiangsu Key Laboratory of Molecular and Functional Imaging, Department of Radiology, Zhongda Hospital, Medical School, Southeast University, Nanjing, China; [4]Key Laboratory of Developmental Genes and Human Diseases, Department of Histology Embryology, Medical School, Southeast University, Nanjing, China; [5]Institute of Basic Medical Sciences Chinese Academy of Medical Sciences, School of Basic Medicine Peking Union Medical College, Beijing, China; [6]Research Unit of Mitochondria in Brain Diseases, Chinese Academy of Medical Sciences, PKU-Nanjing Institute of Translational Medicine, Nanjing, China; [7]Academy of Advanced Interdisciplinary Study, College of Engineering, Peking University, Beijing, China; [8]State Key Laboratory of Membrane Biology, Beijing Key Laboratory of Cardiometabolic Molecular Medicine, Peking-Tsinghua Center for Life Sciences, Institute of Molecular Medicine, College of Future Technology, Peking University, Beijing, China; [9]National Biomedical Imaging Center, State Key Laboratory of Membrane Biology, Peking-Tsinghua Center for Life Sciences, College of Future Technology, Peking University, Beijing, China

*For correspondence:
xiaochun.gu@seu.edu.cn (XG);
chengp@pku.edu.cn (HC)

[†]These authors contributed
equally to this work

Competing interest: The authors declare that no competing interests exist.

**Abstract** Microglia surveillance manifests itself as dynamic changes in cell morphology and functional remodeling. Whether and how microglia surveillance is coupled to brain state switches during natural sleep-wake cycles remains unclear. To address this question, we used miniature two-photon microscopy (mTPM) to acquire time-lapse high-resolution microglia images of the somatosensory cortex, along with EEG/EMG recordings and behavioral video, in freely-behaving mice. We uncovered fast and robust brain state-dependent changes in microglia surveillance, occurring in parallel with sleep dynamics and early-onset phagocytic microglial contraction during sleep deprivation stress. We also detected local norepinephrine fluctuation occurring in a sleep state-dependent manner. We showed that the locus coeruleus-norepinephrine system, which is crucial to sleep homeostasis, is required for both sleep state-dependent and stress-induced microglial responses and $\beta_2$-adrenergic receptor signaling plays a significant role in this process. These results provide direct evidence that microglial surveillance is exquisitely tuned to signals and stressors that regulate sleep dynamics and homeostasis so as to adjust its varied roles to complement those of neurons in the brain. In vivo imaging with mTPM in freely behaving animals, as demonstrated here, opens a new avenue for future investigation of microglia dynamics and sleep biology in freely behaving animals.

## eLife assessment

This **important** study uses cutting-edge miniature two-photon microscopy to follow the structural dynamics of microglia in the somatosensory cortex of freely-moving mice across the sleep/wake

cycle. **Solid** evidence revealed the brain-state-dependent regulation of microglial activity, highlighting alterations in microglial morphology during REM and NREM sleep phases compared to wakefulness. Furthermore, this study provides **important** evidence for a critical role of norepinephrine from the locus coeruleus as a modulator of microglial morphology through the β2-adrenergic receptor (b2AR). Overall, the article is an **exceptional** technical feat to bridge a crucial gap in understanding sleep state-induced dynamics of microglia and its modulation by norepinephrine signaling.

## Introduction

Sleep is a highly conserved and essential phenomenon of the brain and serves numerous cognitive as well as metabolic and immunologic functions. Moreover, the sleep-wake transition represents a specific and prominent change in the brain state (*Besedovsky et al., 2019*; *Rasch and Born, 2013*). As compared to a wakeful brain, the sleeping brain shows more synchronous neuronal activities, which can be characterized by electroencephalography (EEG). Sleep has two main stages: the non-rapid eye movement (NREM) state, which is thought to promote clearance of metabolic waste partly through increased flow of interstitial fluid (*Feng et al., 2019*), and the rapid eye movement (REM) state, with EEG patterns similar to wakefulness yet with muscle atonia as well as vivid dreaming in humans (*Besedovsky et al., 2019*). In addition, sleep, especially NREM, is directly involved in memory consolidation (*Fogel and Smith, 2011*; *Rasch and Born, 2013*). However, the underlying biological dynamics and especially the cellular mechanisms of sleep regulation are not entirely understood (*Frank, 2018a*; *Frank and Heller, 2018b*).

Growing evidence shows that microglia dynamics are coupled to and intertwined with the sleep-wake cycle (*Deurveilher et al., 2021*; *Hristovska et al., 2022*). Microglia are resident innate immune cells ubiquitously distributed in the central nervous system and account for 10–15% of all brain cells (*Wolf et al., 2017*). Microglia surveillance refers to dynamic changes in cell morphology that accompany functional remodeling in response to changes in the neural environment (*Liu et al., 2019*; *Nimmerjahn et al., 2005*; *Stowell et al., 2019*). It has been shown that microglia ablation disrupts the maintenance of wakefulness and promotes NREM sleep (*Corsi et al., 2022*; *Liu et al., 2021*), while enhancing an animal's fear memory consolidation (*Wang et al., 2020*). Conversely, acute and chronic sleep deprivation (SD) leads to microglia activation amidst a proinflammatory state triggered by elevated circulating levels of cytokines (C-reactive protein, TNF-α, IL-1, and IL-6) (*Krueger et al., 2011*). Thus, it is of critical interest to explore whether microglia surveillance can sense and respond to brain state changes during both natural sleep-wake cycles as well as under SD stress.

The investigation of microglial surveillance during the sleep-wake cycle demands technologies suitable to track the rapid dynamics of natural microglia behavior in real-time. Though there have been pioneering studies on microglia dynamics in cultured brain slices (*Honda et al., 2001*), static ex vivo conditions can hardly recapitulate the dynamic microenvironments and neuronal activities in vivo. Based on results obtained from fixed brain slices, it has been shown that chronic sleep restriction but not acute sleep loss causes microglia process changes (*Bellesi et al., 2017*), but it is difficult to use this approach to reconstruct the dynamic behavior of microglia accurately and quantitatively. As a major advance, two-photon microscopy has recently been applied to image microglia in head-fixed or anesthetized animals, revealing microglial process motility, chemotaxis and homeostatic translocation, and multifaceted microglia-neuron interactions in vivo (*Davalos et al., 2005*; *Eyo et al., 2018*; *Liu et al., 2019*; *Stowell et al., 2019*). However, because of mechanical constraints or anesthetics, this technology still precludes experimental paradigms in which the natural sleep-wake cycle is undisturbed. In this regard, the recent advent of mTPM has provided a powerful new tool ideal for fast and high-resolution brain imaging in freely behaving rodents (*Zong et al., 2021*; *Zong et al., 2017*). With the aid of mTPM, researchers have mapped the functional network topography of the medial entorhinal cortex (*Obenhaus et al., 2022*), deciphered the microcircuit dynamics in the dorsomedial prefrontal cortex during social competition (*Zhang et al., 2022*), and unraveled the specific itch signal processing in the primary somatosensory cortex (*Chen et al., 2021*).

In this study, we aimed to determine whether microglia surveillance undergoes any significant changes during natural sleep-wake cycles, and if so, what is the underlying regulatory mechanism. Using different models of mTPM tailored for high-resolution, multi-plane, or large field-of-view (FOV) imaging, we monitored microglia in the somatosensory cortex of mice during natural sleep-wake

cycles or while being subjected to acute SD. We found robust sleep state-dependent and SD-induced changes in microglia surveillance with characteristics differing from those induced by anesthetics. Furthermore, we showed that norepinephrine signals from the axonal projections of the locus coeruleus (LC) underlie the state-dependence of microglia surveillance, and $\beta_2$-adrenergic receptor ($\beta_2$AR) signaling plays a significant role in this process.

## Results

### Imaging microglia dynamics in freely behaving mice

To visualize microglial morphology and dynamics during the sleep-wake cycle, microglia were specifically labeled with a green fluorescent protein (GFP) expressed under the control of an endogenous Cx3cr1 promoter (Cx3cr1-GFP). Time-lapse imaging of the somatosensory cortex utilized the fast and high-resolution miniature two-photon microscope (FHIRM-TPM) recently developed by our laboratory (*Zong et al., 2017*), and EEG/Electromyography (EMG) signals and behavior videos were simultaneously recorded to determine corresponding sleep-wake states (*Figure 1a–c*, *Figure 1—figure supplement 1*). The focal plane of the FHIRM-TPM was placed 50–100 µm beneath the surface of the somatosensory cortex with a FOV of 220 µm × 220 µm (*Figure 1d*). Under the experimental conditions, we were able to track morphological and positional changes of a cohort of 5–10 microglia at a frame rate of five frames per second (FPS) over prolonged durations of greater than 10 hr (*Figure 1—figure supplement 1*, *Figure 1—videos 1 and 2*), while the animals were allowed to roam and behave freely and their sleep-wake cycles remained undisturbed. Though, we were initially concerned with potential phototoxicity associated with prolonged mTPM imaging, no changes in microglial morphology and fluorescent intensity were evident even after non-stop recording over the entire protocol period (*Figure 1—figure supplement 1*). This remarkable result indicates that photodamage is minimal, if any, during in vivo imaging (*Figure 1—figure supplement 1*).

Dynamic changes in microglial area, branching points, and process endpoint speed were then analyzed using Imaris software (*Figure 1 e1–e3'*, *Figure 1—figure supplement 1*). We found that individual microglia maintained a relatively stable territory of surveillance over the entire experimental period, even while their individual projections extended and contracted incessantly, giving rise to a process endpoint motility of 1.39 ± 0.09 µm/min in the wake state (*Figure 1f–h*). These results show that mTPM in conjunction with multi-modal recordings of EEG/EMG and behavioral videos enables in vivo visualization of microglia dynamics over multiple time scales in freely behaving mice. Interestingly, we also observed occasional short-range translocation in a few cells, suggesting the ability of microglia to survey territories beyond their own boundaries, in general agreement with recent observations in head-fixed mice (*Eyo et al., 2018*).

### Microglia surveillance during the sleep-wake cycle

By simultaneously imaging microglia and recording EEG/EMG throughout a sleep-wake cycle, we obtained continuous 4–6 hr datasets from different mice and, in conjunction with video interpretation, segmented and sorted them into subgroups corresponding to wake, NREM, and REM states (*Figure 2a*). For each state, we quantified morphological features and process motility from 15 to 20 cells. Wake and NREM states lasting longer than 1 min were identified, for which the last 30 s recording was selected, and REM states lasting longer than 30 s for the characterization of possible state-dependent microglia dynamic surveillance.

We found that during the NREM and REM states, microglial process length, and surveillance area increased compared with those during wakefulness (*Figure 2b–d and b'–d'*). Concomitantly, the number of branching points significantly increased, while endpoint moving speed of microglial processes decreased (*Figure 2e–h*), indicating state-dependent changes in microglial morphology as well as process motility under natural sleep-wake cycles. To provide a volumetric view of the dynamics of microglial morphology, we adapted our latest version of mTPM with an electrical tunable lens (ETL) capable of imaging variable focal planes (*Figure 2i–o*, *Figure 2—figure supplement 1*; *Zong et al., 2021*). Multi-plane reconstruction revealed more extended process lengths and greater branch point numbers as compared to single-plane imaging as expected (*Figure 2l and n*). The volume of surveillance for individual microglia changed from 1661 ± 264.4 µm³ in the wake state to 3802 ± 670.5 µm³ in REM state and 4616 ± 324.6 µm³ in NREM state (*Figure 2m*). Similar to results obtained with

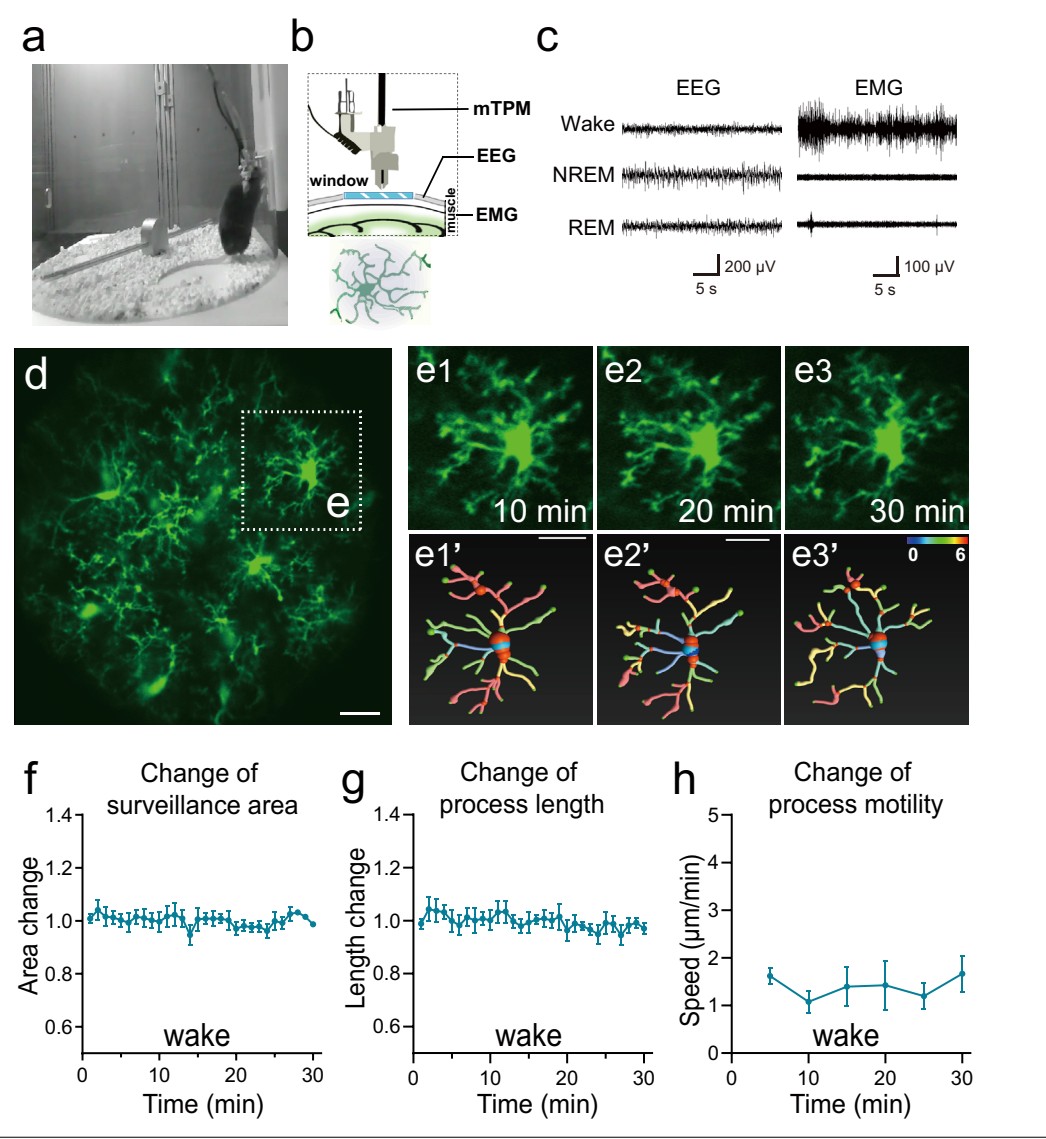

**Figure 1.** Imaging microglial surveillance in the somatosensory cortex in freely moving mice. (**a–c**) Experimental setup. The animal was head-mounted with a miniature two-photon microscope (mTPM) and electroencephalography (EEG)/electromyography (EMG) electrodes and behaved freely in a cylindrical chamber (**a**). Microglia expressing green fluorescent protein (GFP) in the somatosensory cortex were imaged through a cranial window using the mTPM (**b**) and the sleep/wake state of the animal was simultaneously monitored using an EEG/EMG recording system (**b–c**). (**d-e**) Microglial morphological dynamics when the animal was awake. A representative image with a field-of-view (FOV) of 220 μm × 220 μm. (e1–e3) Expanded views of selected microglia in box from (**d**) at 10 min (e1), 20 min (e2), and 30 min (e3) of continuous recording. (e1'-e3') Microglial process graphs digitally reconstruction for e1-e3 using Imaris software. In Figure e1'-e3', the branch points of processes are represented in red and the number of different branches of the whole cell is represented as gradient colors. (**f–h**) Quantitative analysis of changes in microglia surveillance area (**f**), process length (**g**), and process motility, indexed by the speed of the extension and retraction at endpoints of the processes (**h**). Note that the gross morphology of microglia remained largely unchanged over a 30 min time frame in the wake state, despite significant motility at the ends of the processes. Scale bars, 25 μm. n=12 cells from three mice.

The online version of this article includes the following video and figure supplement(s) for figure 1:

**Figure supplement 1.** Microglia dynamics imaging and analysis in freely behaving mice.

**Figure 1—video 1.** Imaging data of a 1 hr continuous recording of somatosensory cortex in a freely behaving mouse.

https://elifesciences.org/articles/86749/figures#fig1video1

*Figure 1 continued on next page*

*Figure 1 continued*

**Figure 1—video 2.** Correction for xy motion artifacts (field-of-view of 220 μm × 220 μm).
https://elifesciences.org/articles/86749/figures#fig1video2

single-plane imaging, the endpoint speed of processes decreased in REM and NREM compared to the wake state (*Figure 2o*). While both REM and NREM had the same trends in change as compared to the wake state, significantly greater changes were associated with NREM than REM in terms of cell volume, branch length, branch number, and endpoint speed (*Figure 2l–o*). These results demonstrate differential microglial dynamic surveillance as the brain state switches among wakefulness, REM and NREM, and the state-dependent morphological and process-motility changes can be robustly reproduced with either single-plane or volumetric imaging. Intriguingly, microglia can even sense and respond to subtle differences between REM and NREM states. Because more extended microglia are thought to have greater surveillance ability (*Kierdorf and Prinz, 2017*), our results support the notion that microglia during the NREM and REM periods tend to be more active in the opportune clearing of metabolic wastes accumulated during the wake states.

To this end, it is also instructive to compare and contrast REM/NREM state-dependent characteristics with those induced by anesthesia. In mice under isoflurane (1.2% in air) for 20–30 min, we found that microglial length, area, and number of branching points (*Figure 2—figure supplement 1*) were similarly increased compared with the wake state, as was the case in NREM and REM. However, the endpoint movement of processes exhibited opposite changes (*Liu et al., 2019*), increasing with anesthesia but decreasing in sleep states (*Figure 2h and o*; *Figure 2—figure supplement 1*). This result shows that microglia surveillance appears to differ qualitatively between sleep and anesthesia states.

## Acute SD promotes the contraction of microglial processes

Sleep loss is detrimental to brain function and results in structural plastic changes in nerve cells. Whether microglia dynamic surveillance alters after sleep loss remains controversial (*Bellesi et al., 2017*; *Wang et al., 2020*). Therefore, we induced acute SD in mice by forcing the animals to exercise and interrupting sleep with a rotatory rod, starting at 9:00 AM and lasting for 6 hr (zeitgeber time 2–8) (*Figure 3a*). The duration of recovery sleep, especially NREM sleep, increased after SD (*Figure 3—figure supplement 1*). In the wake state, microglia typically exhibited an extended form with ramified, long, thin processes. Overt morphological changes occurred as early as 3 hr after SD, when microglia presented a contracted form bearing short and thick processes. By 6 hr, nearly all microglia cells were converted into a phagocytic form, assuming an amoeboid shape with few processes (*Figure 3b–d*). We found that microglial length, area, and number of branching points decreased with the continuous SD (*Figure 3e–h*). The same changes were observed in 3D reconstruction (*Figure 3—figure supplement 1*). Changes in the endpoint speed of processes appeared to be biphasic, increasing at 3 hr SD and then declining below baseline level at 6 hr SD (*Figure 3h*); the latter may result in part from the dramatic shrinkage of the processes. As the sleep pressure was eased during recovery sleep, morphological changes of microglia were partially reversed over a timescale of several hours (*Figure 3—figure supplement 1*). Thus, our real-time recording in live animals clearly demonstrates that SD stress induces microglia to assume an active state, and conspicuous microglial response occurs early in SD.

## NE changes in the somatosensory cortex during the sleep-wake cycle

Next, we sought to determine a possible signaling mechanism(s) that incites microglia surveillance state- and stress-dependent changes. Among a number of candidates responsible for stress responses, norepinephrine (NE) has recently been shown to play a crucial role in microglia surveillance by responding to neuronal network activity (*Liu et al., 2019*) and partaking of synaptic plasticity (*Stowell et al., 2019*). Therefore, we combined mTPM imaging with EEG and EMG recording to determine whether NE levels in the somatosensory cortex fluctuate during the sleep-wake cycle to control microglia surveillance. We imaged extracellular NE dynamics in somatosensory cortical neurons with NE biosensor GRAB$_{NE2m}$ (*Figure 4a*), which was developed based on G protein receptors (*Feng et al., 2019*; *Kjaerby et al., 2022*). The biosensor was expressed on the plasma membrane of neurons to report the dynamic change of the extracellular NE. Dynamic changes in NE levels were evident in the somatosensory cortex during wake, NREM, and REM states (*Figure 4b*). NE levels reached higher levels during longer awakenings, while the lowest NE levels were detected during

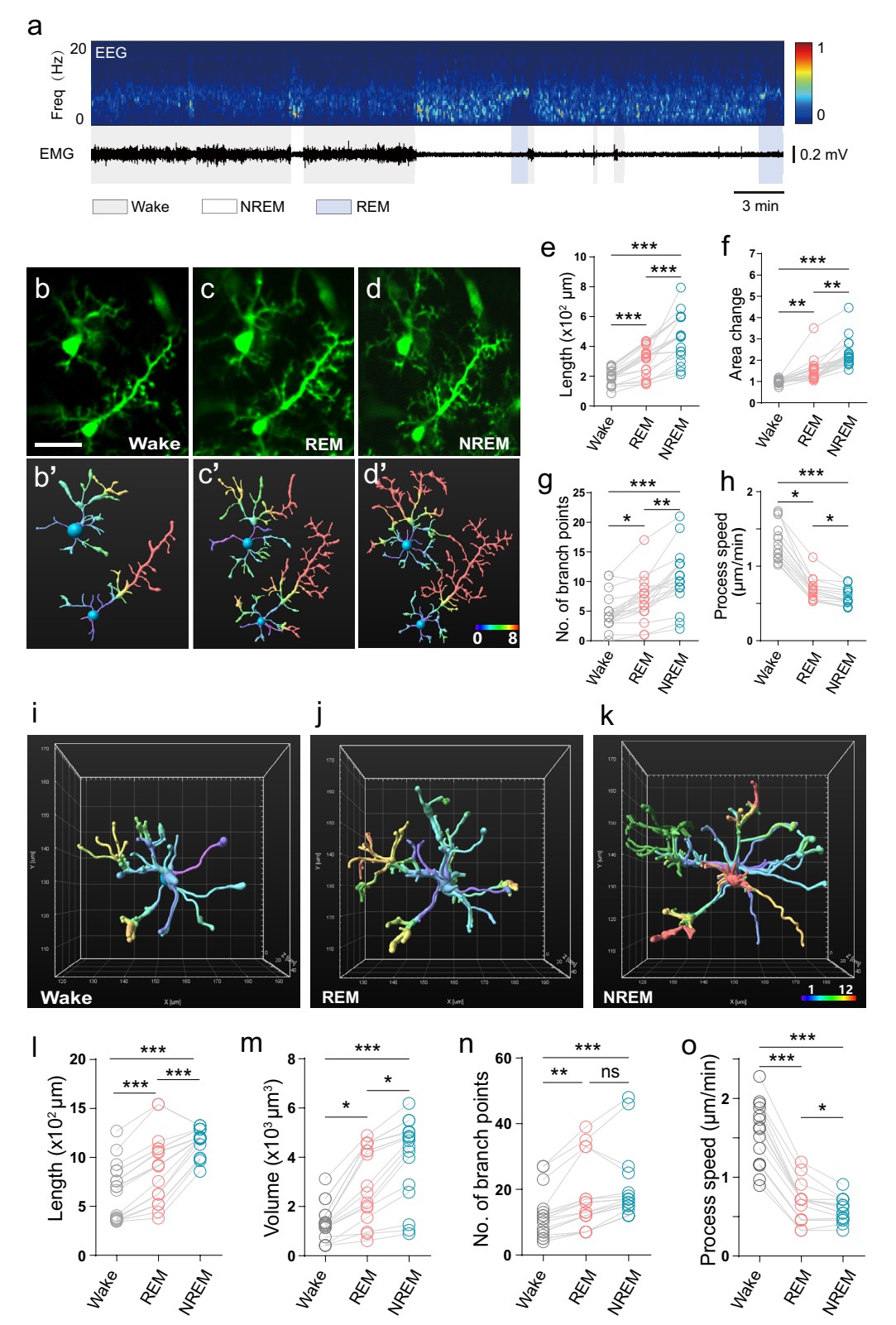

**Figure 2.** Microglial surveillance is state-dependent in the sleep-wake cycle. (**a**) Representative electroencephalography (EEG)/electromyography (EMG) recordings showing the sleep-wake stage switch. Top, EEG power spectrogram (0–20 Hz). Middle, EMG trace. Bottom, brain states are classified as wake (color code: gray), rapid eye movement (REM) (blue), and non-rapid eye movement (NREM) (white). (**b–d**) Representative microglial morphological changes during the sleep-wake cycle. (**b'-d'**) Microglial morphology reconstructed from b-d using Imaris software. (**e–h**) Microglial morphological

*Figure 2 continued on next page*

*Figure 2 continued*

parameters, length (**e**), area change (**f**), number of branch points (**g**), and process endpoint speed (**h**), all exhibited brain state-dependent dynamic change. One-way ANOVA with Tukey's post-hoc test in **e**; Friedman test with Dunn's post-hoc test in **f–h**; n=20 cells from seven mice for each group (**e–g**), n=15 cells from six mice for each group (**h**); *p<0.05, **p<0.01, ***p<0.001. (**i–o**) 3D multi-plane imaging and reconstruction of microglial morphology. A 3D electrical tunable lens (ETL) lens was used to acquire multi-plane imaging (220 μm × 220 μm × 40 μm) at Z-intervals of 2 μm, at a rate of 7.5 stacks/5 min. (**i–k**) 3D reconstructed microglial morphology in wake (**i**), REM (**j**), and NREM (**k**) states, with corresponding time stamps shown at the bottom. (**l–o**) Quantitative analysis of microglial length (**l**), volume change (**m**), number of branch points (**n**), and process motility (**o**) based on multi-plane microglial imaging. Data from 3D imaging corroborated state-dependent changes of microglial morphology in the sleep-wake cycle. Scale bars, 30 μm. One-way ANOVA with Tukey's post-hoc test in **l, o**; Friedman test with Dunn's post-hoc test in **m, n**; n=17 cells from 7 mice for each group (**l-n**), n=15 cells from six mice for each group (**o**); *p<0.05, **p<0.01, ***p<0.001.

The online version of this article includes the following figure supplement(s) for figure 2:

**Figure supplement 1.** Multiplane imaging of microglial surveillance and changes of microglial surveillance under anesthesia.

REM sleep (*Figure 4b*), and the largest changes in NE levels were recorded occurred during the transition from REM to wake (*Figure 4b*). During NREM sleep, the NE oscillated dynamically around a relatively low level (*Figure 4b*), with brief increases that might be related to memory consolidation (*Kjaerby et al., 2022*). On average, mean levels of NE exhibited a brain-state-dependent change, varying in a decreasing order from wake to NREM and to REM (*Figure 4c*). These results are in general agreement with recent observations that NE fluctuates in a sleep-state-dependent manner in the medial prefrontal cortex, and suggest that NE may play a crucial role in controlling the stage-dependent microglial changes.

## Role of the LC-NE signal in controlling the microglia dynamic surveillance

To determine possible involvement of NE in sleep state- and SD-induced microglia surveillance, we used the LC-selective neurotoxin, N-(2-chloroethyl)-n-ethyl-2-bromobenzylamine (DSP4; applied two days before imaging) to ablate LC axons projecting into the cortex (*Figure 5—figure supplement 1*; *González et al., 1998*) and characterized microglial surveillance with and without an LC-NE signal (*Figure 5a*). DSP4 application led to a significant increase in sleep states, particularly NREM sleep (*Figure 5—figure supplement 1*), in agreement with previous studies (*González et al., 1998*). Meanwhile, the characteristics of NE dynamics in different brain states were also changed by DSP4 (*Figure 5—figure supplement 1*). Importantly, the sleep state-dependent changes in microglia surveillance were abolished altogether; neither microglial morphology nor process motility displayed any of the previously significant changes seen when the brain states cycled across wake-REM-NREM (*Figure 5b–g*). Likewise, DSP4 treatment completely prevented SD-induced alteration of microglia surveillance (*Figure 5—figure supplement 1*). These results indicate an essential role for the LC-NE signal in sleep- and stress-dependent state modulation of microglia surveillance.

The effects of the neural modulator NE are mediated by two families of G-protein-coupled receptors, α and β-adrenergic receptors (ARs), each comprising several subtypes and it has been shown that $\beta_2AR$ stimulation rapidly induces microglia dynamic surveillance (*Gyoneva and Traynelis, 2013*; *Liu et al., 2019*; *Stowell et al., 2019*). We, therefore, examined microglia surveillance in $\beta_2AR$ KO animals during natural sleep-wake cycles and under SD stress. Similar to LC axon ablation, $\beta_2AR$ KO disturbed wake-sleep homeostasis, as manifested by a lengthening of overall sleep duration (*Figure 5—figure supplement 1*). By contrast, the removal of $\beta_2ARs$ failed to abolish dynamic changes in microglia when the brain state switched between sleep and wakefulness (*Figure 5h–n*). Nonetheless, the ability of microglia to subtly distinguish between REM and NREM states was largely compromised (*Figure 5h–n*, *Figure 5—figure supplement 1*), suggesting a partial contribution of $\beta_2AR$ signaling to LC-NE modulation of microglia surveillance during sleep-wake cycles. Regarding microglia responses to SD stress, we showed that microglial morphology and motility remained unchanged during the SD protocol in $\beta_2AR$ KO animals (*Figure 5—figure supplement 1*), revealing a predominant role for $\beta_2AR$ signaling in LC-NE modulation of microglia surveillance under SD stress.

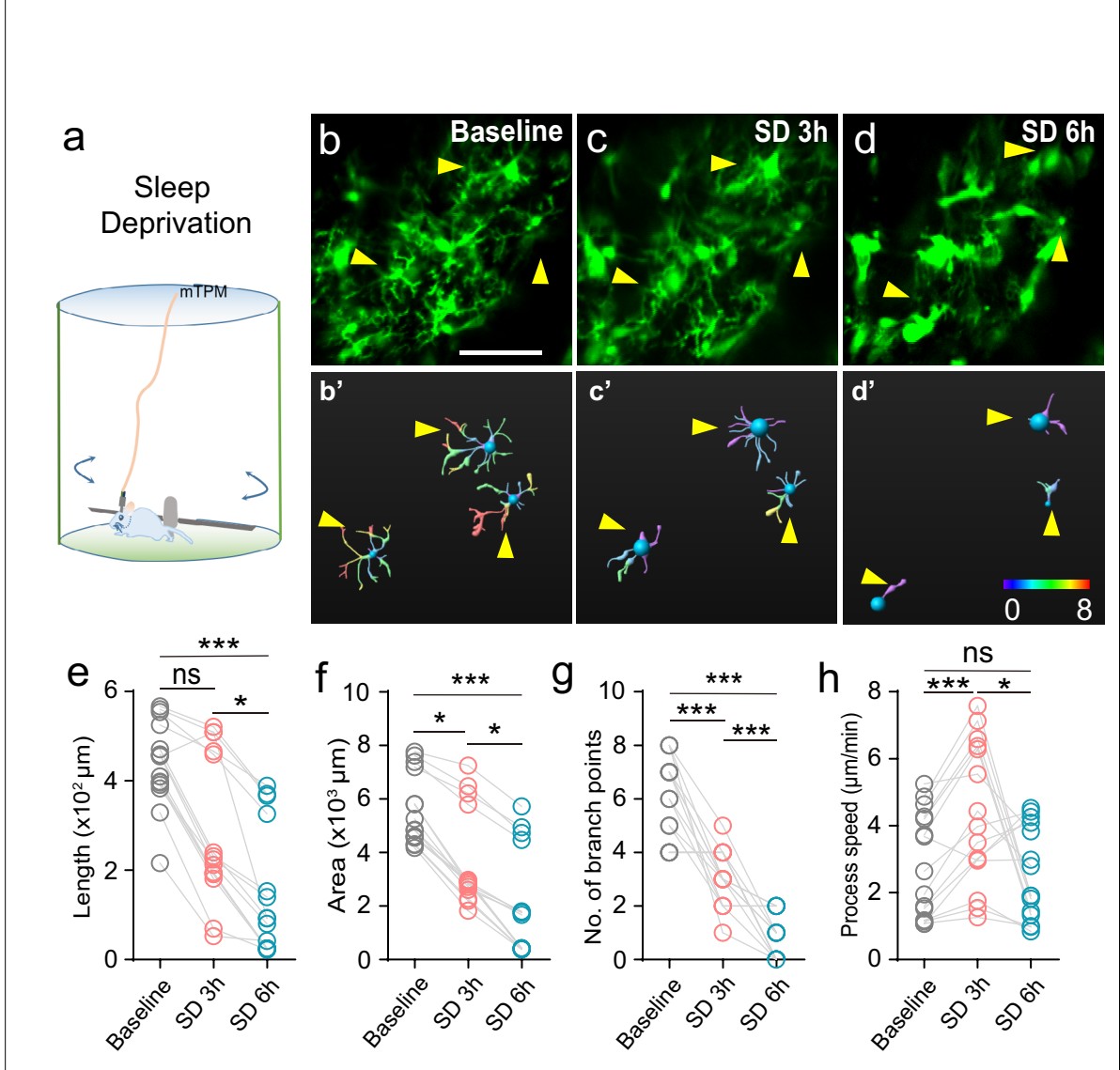

**Figure 3.** Changes of microglial surveillance in the state of sleep deprivation. (**a**) Experimental setup for sleep deprivation. Sleep deprivation in mice was achieved by forcing them to exercise and interrupting their sleep with the rotation of a 46 cm rod (18 turns/min) in the chamber (diameter 50 cm). (**b–d**) Microglial processes contracted after sleep deprivation (SD), baseline (**b**), SD 3 hr (**c**), and SD 6 hr (**d**). (**b'-d'**) Morphological changes of microglia reconstructed using Imaris software. **b'-d'** correspond to **b-d**, respectively. (**e–h**) Statistics for length (**e**), area (**f**), number of branch points (**g**), and process motility (**h**). Scale bars, 30 μm. One-way ANOVA with Tukey's post-hoc test in **g, h**; Friedman test with Dunn's post-hoc test in **e, f**; n=15 cells from six mice for each group; *p<0.05, ***p<0.001.

The online version of this article includes the following figure supplement(s) for figure 3:

**Figure supplement 1.** Changes of microglial surveillance in the state of sleep deprivation and recovery.

## Discussion

With the development of mTPM imaging and EEG/EMG recording in freely behaving mice, we now provide direct evidence that microglia surveillance in mouse somatosensory cortex is sleep-state dependent during natural sleep-wake cycles and that microglia contract from ramified into phagocytic forms a few hours after the onset of acute SD stress. Combined with pharmacological intervention and the use of NE biosensors and genetically manipulated animals, we showed that the LC-NE pathway mediates these state-dependent changes in microglia surveillance and that β₂AR signaling is involved in various aspects of the microglial responses, extending recent reports that the same

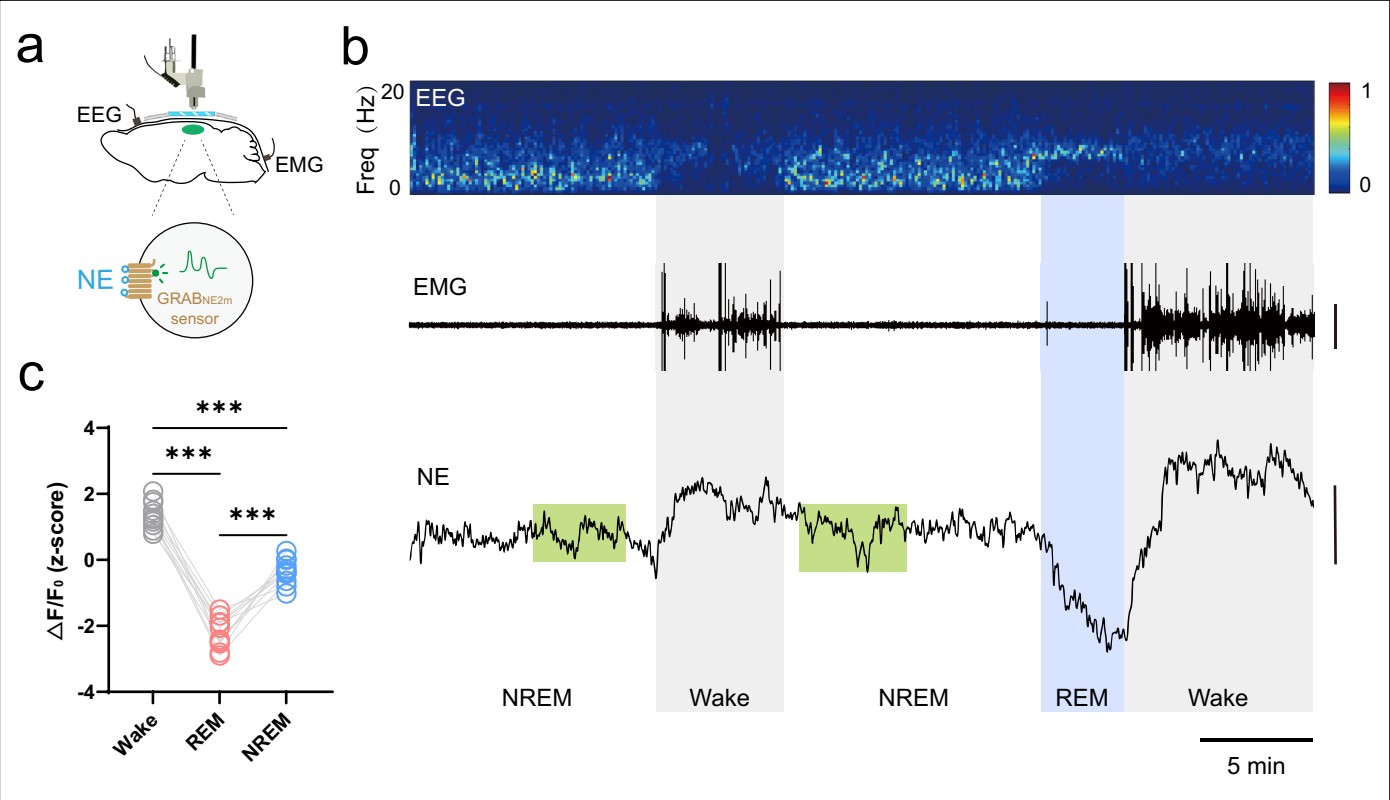

**Figure 4.** Norepinephrine (NE) dynamics in mouse somatosensory cortex during the sleep-wake cycle. (**a**) Schematic diagram depicting miniature two-photon microscopy (mTPM) recording of extracellular NE indicated by the GRAB_NE2m sensor expressed in neurons. (**b**) Representative traces of simultaneous recordings in the somatosensory cortex during the sleep-wake cycle in freely behaving mice. Electroencephalography (EEG) and its power spectrogram (0–20 Hz); electromyography (EMG) (scale, 50 μV); NE signals reflected by the z-score of the GRAB_NE2m fluorescence (scale, 2 z-score). The brain states are color-coded (wake, gray; non-rapid eye movement (NREM), white; REM, blue; NE oscillatory during NREM sleep, green). (**c**) Mean extracellular NE levels in different brain states. Data from the same recording are connected by lines. ***$p<0.001$, one-way ANOVA with Tukey's post-hoc test (n=11 from three mice).

pathway underlies microglia surveillance in the context of neuronal network activity (*Liu et al., 2019*) and synaptic plasticity (*Stowell et al., 2019*).

That microglia surveillance is sleep state-dependent substantiates an intimate relationship between rapid, robust state-dependent microglia remodeling and the biology of the sleep-wake cycle (*Feng et al., 2019*; *Frank and Heller, 2018b*; *Xie et al., 2013*). It has been shown that microglia play a central role in the removal of metabolic waste and even cell debris from the brain (*Huisman et al., 2022*; *Márquez-Ropero et al., 2020*). Our study shows that microglia in the sleeping brain has a larger area and length, consistent with greater clearance ability in the sleeping state than the waking. Recently, it has been reported that, in head-fixed mice, microglial morphological complexity was decreased in NREM sleep than during wakefulness (*Hristovska et al., 2022*). In consideration of the altered durations of both NREM and REM sleep (*Hristovska et al., 2022*), the unnatural sleep state would lead to an increase in the microarousal state, and ultimately lead to a change in the structure of the sleep state, which may be the main reason for the difference in microglia behavior from our natural sleep. Furthermore, our results show that microglia surveillance significantly differs between REM and NREM sleep in the somatosensory cortex, likely reflecting metabolic and functional differences between the two states. More specifically, previous reports have proposed differential roles between the NREM and REM sleep for the consolidation of different types of memories. For example, NREM sleep contributes to declarative memories, whereas REM sleep is important for procedural and emotional memories (*Besedovsky et al., 2019*; *Rasch and Born, 2013*). Memory consolidation is closely related to synaptic plasticity, in which microglia play a very important role (*Corsi et al., 2022*; *Tuan and Lee, 2019*). In this regard, our findings hint at the possibility that not all features of

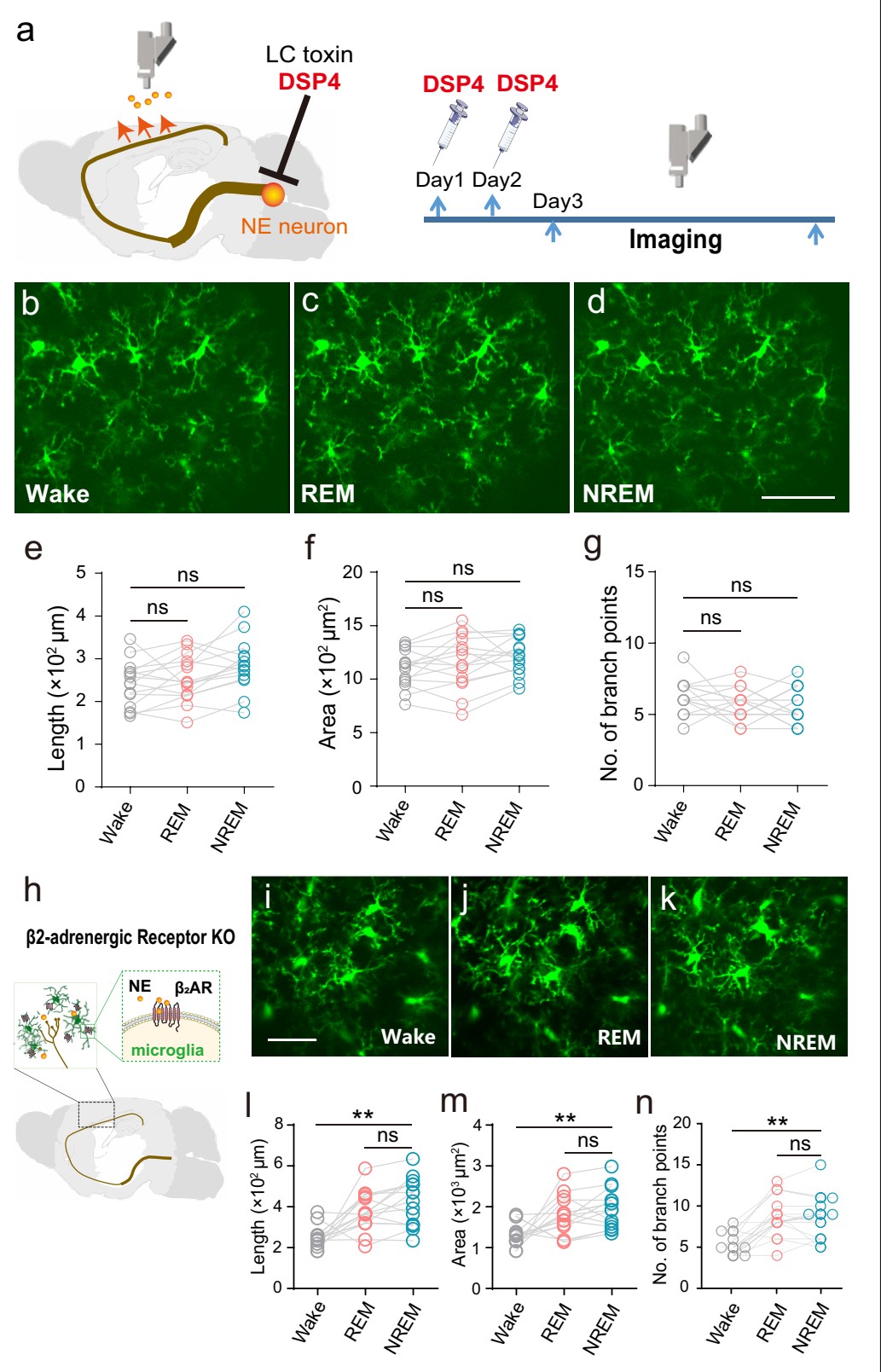

**Figure 5.** Microglial surveillance during natural sleep is controlled by LC-NE signal. (**a**) Experimental setup: LC-selective neurotoxin DSP4 was used to destroy LC-NE neuronal axons. (**b–g**) Lack of sleep/wake state-dependent microglial surveillance in LC- axon ablated animals. b-d: Representative miniature two-photon microscopy (mTPM) images of microglia at different states. e-g: Statistics for microglial length (**e**), surveillance area (**f**), and number of

*Figure 5 continued on next page*

*Figure 5 continued*

branch points (**g**) in LC-axon ablated mice. One-way ANOVA with Tukey's post-hoc test in **e, f**; Friedman test with Dunn's post-hoc test in **g**; n=15 cells from six mice for each group; ns, not significant. (**h**) Schematic diagram for $\beta_2$ARs on the plasma membrane of microglia in the cerebral cortex responding to norepinephrine (NE) released from axonal terminals projected from locus coeruleus (LC). (**i–k**) State-dependent microglial surveillance during sleep-wake cycle in $\beta_2$AR knockout mice. Representative microglial images (**i–k**) and statistics for microglial process length (**l**), surveillance area (**m**), and number of branch points (**n**) at different states in (CX3CR1-GFP+/-; Adrb2-/-) mice. Scale bars, 30 µm. One-way ANOVA with Tukey's post-hoc test in **m**; Friedman test with Dunn's post-hoc test in **l, n**; n=15 cells from six mice for each group; ns, not significant, **$p<0.01$.

The online version of this article includes the following figure supplement(s) for figure 5:

**Figure supplement 1.** Altered sleep-wake states after DSP4 administration and $\beta_2$AR knockout and controlling microglial surveillance during sleep deprivation (SD) by LC-NE signal.

microglia-neuron interactions, including mode of action and localization, are equal in the consolidation of different types of memory.

It has been shown that a large proportion of Iba1-immunoreactive microglia with larger cell bodies and less ramified processes appear in hippocampal brain slices following 48–72 hr of sleep deprivation and in the frontal cortex after 4.5 days of chronic sleep restriction, whereas acute sleep deprivation for 8 hr had no effect on microglial morphology (*Bellesi et al., 2017*; *Hall et al., 2020*). However, these findings have mostly been derived from slice staining. While the present work supports the general conclusion that microglia surveillance is responsive to SD stress, our real-time mTPM imaging in vivo also unmasked a progressive change in microglia surveillance that occurred as early as 3 hr after onset of SD and full conversion of microglia into the phagocytic form after 6 hr SD stress. This apparent disparity between the current and previous reports might reflect the complications arising from ex vivo sample preparation, because microglia surveillance is highly dynamic and sensitive to neural environmental changes, evidenced by its cyclic changes during the sleep-wake switch.

We have provided two lines of evidence that the LC-NE-$\beta_2$AR axis is involved in microglial surveillance both during the natural sleep-wake cycle and under SD stress. Ablation of LC-NE neuronal projection abolished and knockout of $\beta_2$ARs markedly altered the manifestation of the state-dependence of microglia surveillance, while the latter recapitulated most, but not all, the phenotypes of the former. These findings are well supported by emerging evidence that NE is necessary for microglial activation (*Bellesi et al., 2016*; *Berridge et al., 2012*; *Mori et al., 2002*). First, NE is known to be a key neurotransmitter that regulates sleep dynamics (*Wang et al., 2020*) and the dynamics of microglia (*Liu et al., 2019*; *Sugama et al., 2019*). Second, it has been shown that $\beta_2$AR expressed at a higher level in microglia than other CNS cell types (*Zhang et al., 2014*) and microglia dynamic surveillance rapidly responds to $\beta_2$AR stimulation (*Gyoneva and Traynelis, 2013*; *Liu et al., 2019*; *Stowell et al., 2019*). Moreover, the morphology of microglia in the somatosensory cortex is also regulated by neuronal activity, in a microglial $\beta_2$AR-dependent manner (*Liu et al., 2019*). Our results also extend the classic view of NE regulation of sleep and wakefulness by revealing a prominent microglial component in the NE response. It should be noted that the pan-tissue $\beta_2$AR knockout animal model was used in the current study and it warrants future investigation to pinpoint specific roles of microglial $\beta_2$AR in the brain-state dependent microglial responses.

In addition to NE, other potential modulators also present dynamic during the sleep-wake cycle and may partake in the regulation of microglia dynamic surveillance. It has been reported that LC firing stops (*Aston-Jones and Bloom, 1981*; *Rasmussen et al., 1986*), while inhibitory neurons, such as PV neurons and VIP neurons, become relatively active during REM sleep (*Brécier et al., 2022*). ATP level in the basal forebrain is shown to be higher in REM sleep than NREM sleep (*Peng et al., 2023*). Similarly, the adenosine level in the somatosensory cortex during REM sleep is higher than in NREM sleep (Response *Figure 1*). These considerations may explain the finding that $\beta_2$AR knockout failed to abolish microglial responses to sleep state switch and SD stress altogether. A more complete understanding of the regulatory mechanisms of microglia surveillance will help to delineate more precisely the roles of microglia in physiology and under stress.

In summary, by real-time in vivo mTPM imaging of microglia surveillance, we uncovered the state-dependence of microglia surveillance during natural sleep-wake cycles and a robust, early-onset response to SD stress. Both types of microglial dynamics are under the regulation of LC-NE with the

involvement of $\beta_2$ARs as one of its main effectors. These results highlight that microglia undergo rapid and robust remodeling of both morphology and function to fulfill multimodal roles complementary to those of neurons in the brain. In addition, the methodologies established in the present study may prove to be of broad application for future investigations of microglia dynamics and sleep biology in freely behaving animals.

## Methods

### Animals

All experimental protocols were carried out with the approval of the Institutional Animal Care and Use Committee of PKU-Nanjing Institute of Translational Medicine (Approval ID: IACUC-2021–023). Male mice, 2–3 months of age, were used in accordance with institutional guidelines. Cx3cr1-GFP (Jax, #021160) heterozygous mice were used to visualize microglia with miniature two-photon microscopy (mTPM). To ablate LC-NE neurons, DSP4 (Sigma, #C8417) solution was administered intraperitoneally (50 mg/kg) twice at 24 hr intervals and a minimum of 48 hr before mTPM imaging. Adrb2 KO (Jax, #031496) mice were crossed with Cx3cr1-GFP mice to generate (Cx3cr1-GFP+/-; Adrb2-/-) mice, which were used in experiments with $\beta_2$-adrenergic receptor knock-out. Mice were housed with a standard 12 hr light/12 hr dark cycle and fed standard chow ad libitum.

### Surgery and electrode implantation

All surgical procedures were done sterilely, and all animal-administered reagents were sterile. Mice were anesthetized with isoflurane (1.5% in air at a flow rate of 0.4 L/min) and maintained on a 37 °C heating pad during surgery. The cerebral cortical region to be imaged was localized based on the stereotactic coordinates (somatosensory area, from bregma: anteroposterior, –1 mm, mediolateral, +2 mm) and marked with a fine black dot. The skull over the region of interest was thinned with a high-speed micro-drill under a dissection microscope. Drilling and application of normal saline were done intermittently to avoid overheating and hurting the underlying cerebral tissue. After removing the external layer of the compact bone and most of the spongy bone layer with the drill, the area continued to thinned until an expanded smooth area (~3 mm in diameter) was achieved, making sure the middle was thin enough to get high-quality imaging. A drop of saline was applied and a 3-mm-diameter glass coverslip previously sterilized in 70% ethanol was placed on this window, then dental cement was applied around the glass coverslip. The mice were used in the experiment after 4 weeks of postoperative recovery.

500 nL of AAV9-hSyn-GRAB$_{NE2m}$ (BrainVTA, #PT-2393) were injected into the somatosensory cortex (A/P, –1 mm; M/L, +2 mm; D/V, –0.3 mm) at a rate of 50 nL/min. 3-mm-diameter glass coverslip were implanted and fixed on the somatosensory cortex. The mice were used in the experiment one month after postoperative recovery.

Epidural screw electrodes (diameter 0.8 mm) were implanted bilaterally on the opposite sides of the imaging window for constant EEG recording from the frontal (bregma: anteroposterior, +1.5 mm, mediolateral, +1.5 mm) and parietal cortex (anteroposterior, –2 mm; mediolateral, +2.5 mm). Electrodes were fixed to the skull with dental cement. Another pair of electrodes was inserted into the neck muscles for EMG recording. Then, a headpiece baseplate was attached to the skull with cyanoacrylate and reinforced with dental cement.

### Simultaneous imaging and recording

After recovery from surgery, mice with a clear cranial window and typical EEG/EMG signals were selected for further experiments. With the mice head-fixed on the imaging stage, an mTPM imaging stack was first acquired using an mTPM (Transcend Vivoscope), and a proper FOV with 5–10 microglial somata was located 50–100 μm beneath the pial surface. The holder of the mTPM was then sealed onto the baseplate over the coverslip on the head, and the mTPM could be repetitively mounted and dismounted to track the same population of microglia over different days (*Figure 1—figure supplement 1*).

Three models of mTPM were used for different experiments: FHIRM-TPM high-resolution model (*Zong et al., 2017*) with lens NA of 0.7 and FOV of 220 μm × 220 μm for *Figures 1–5*; headpiece weight was 2.13 g, and lateral and axial resolutions were 0.74 and 6.53 μm, respectively. FHIRM-TPM

large FOV model (*Zong et al., 2021*) with a headpiece weight of 2.45 g, lens NA of 0.5, and FOV of 420 μm × 420 μm for *Figure 5—figure supplement 1*, and lateral and axial resolutions were 1.13 and 12.2 μm, respectively; FHIRM-TPM 2.0 ETL model *Zong et al., 2021* in *Figure 2—figure supplement 1*, with headpiece weight of 4.3 g, Z-depth range of 45 μm, lens NA of 0.7, lateral resolution of 0.74 μm, axial resolution of 6.53 μm, and FOV of 220 μm × 220 μm. Images were acquired to visualize the microglial surveillance: time-lapse xy imaging stack at a frame rate of 5 FPS or multi-plane imaging stacks (20 planes at 2 μm intervals, five frames/s). Each model of mTPM was equipped with a water-immersion miniature objective and a 920 nm femtosecond fiber laser. Excitation for fluorescence imaging was achieved with 150-fs laser pulses (80 MHz) at 920 nm for GFP with a power of ~25 mW after the objective.

The EEG/EMG signals were recorded continuously and simultaneously by Vital Record (Kissei Comtec system), and mouse behavior was monitored with an infrared video camera. Mice were never disturbed when they were undergoing spontaneous sleep-wake cycles, feeding, or drinking.

### Sleep deprivation

SD was achieved by forcing the mice to move continuously through power devices (Soft maze, #XR-XS108) (*Pandi-Perumal et al., 2007*). The experimental device consisted of a computer console, a horizontal rotating rod with a diameter of 46 cm, and a round mouse cage of 50 cm. The rotating rod can rotate horizontally randomly (three times/min) in clockwise and counterclockwise directions under the control of the computer. The mouse was placed in the cage 1 hr per day for one week for habituation. After baseline sleep recording, mice were sleep-deprived for the duration indicated in each experiment, starting at 9:00 AM and lasting up to 6 hr into SD.

### Anesthesia experiment

Time-lapsed image stacks acquired between 20 and 30 min after anesthesia (isoflurane, 1.5% in air at a flow rate of 0.4 L/min) were used to evaluate microglial dynamic surveillance under anesthetized conditions.

### Quantification and statistical analysis

#### EEG/EMG data analysis

EEG and EMG signals were amplified and filtered as follows: EEG, high-pass filter at 0.1 Hz, low-pass filter at 35 Hz; and EMG, high-pass filter at 10 Hz, low-pass filter at 100 Hz. All signals were digitalized at 128 Hz and stored on a computer. As described previously (*Bellesi et al., 2013*), EEG power spectra were computed by a fast Fourier transform routine for 10 s epochs. Wake, nonrapid eye movement (NREM) sleep, and REM sleep were manually scored off-line (SleepSign, Kissei Comtec) in 10 s epochs according to standard criteria (*Fogel and Smith, 2011*). Epochs containing artifacts, predominantly during animal movement, were excluded from spectral analysis.

#### Microglial morphological analysis

In order to accurately quantify microglial morphometric changes in different brain states, wake, and NREM states lasting longer than 1 min were chosen, among which the last 30 s of recording were selected, in conjunction with video interpretation, for further processing and analysis. Because they were of relatively short duration, REM states lasting longer than 30 s were used for analysis.

Image stacks were first processed offline using ImageJ software to correct for xy motion artifacts, and the time-lapse stack was aligned using the StackReg plugin. In some experiments, motion artifacts due to Z-level drift were further minimized by collecting multi-plane images over ±20 μm depth at 2 μm intervals and then compressing them into a single xy projection to substitute for time-lapsed xy image. In this case, corresponding projections were generated over a 30 s window to substitute for time-lapsed xy images. A maximum-intensity projection (MIP) was then created from the stack for morphological analysis. The criteria to select microglia for analysis included a clearly identifiable soma and processes seen in the x-y plane.

Next, microglial morphological analysis was done using Imaris 9.5 software and the Filament module (*Figure 1—figure supplement 1*). The filament created in Imaris is a connected graph-based object consisting of vertices that are connected by edges. Feature extraction by the Filament module yielded microglial features and parameters defined as the following:

### Microglial process

Same as filament dendrite in the module. Microglial processes form the main structure of the filament object. Processes extend from the microglial cell body and can undergo branching. A process graph is a sequence of vertices connected by edges.

### Number of branch points

Same as filament branch depth in the module. The process graph is a tree-based structure that has a root point. Branching depth is defined as the number of branches or bifurcations in the shortest path from the beginning point to a given point in the process graph.

### Process length

Same as filament dendrite length in the module, in reference to the sum of all process lengths within the entire microglial cell.

### Microglial surveillance area

Same as filament dendrite area in the module, in reference to the sum of all process areas within the entire microglial cell.

### Microglial surveillance volume

Same as filament dendrite volume. In volumetric imaging, it is the sum of the volumes of all processes within the microglial cell.

### Process endpoint speed

Same as filament track speed. The speed is calculated using the 'Track over time' function and given by the absolute track length change, whether extension or retraction, divided by the corresponding elapsed time.

### NE signals analysis

To analyze NE signals data, we binned the raw data into 1 Hz and subtracted the background autofluorescence. $\Delta F/F_0$ were calculated by using the fluorescence signal itself. $F_0$ was defined as the mean value of the lowest fluorescence signal for each 60 s window in each recorded fluorescence signal. The z-score transformed $\Delta F/F_0$ was used for analysis. In order to accurately quantify the changes of NE signals in different brain states, wake and NREM states with a duration of more than 1 min were selected, in which the mean value of 30 s fluorescence signal at the end was selected as the NE signal in this state, which was further processed and analyzed combined with video interpretation. Since the duration of REM state is relatively short and it enters a low level at the end of REM, the mean value of the fluorescence signal in the 30 s before the transition from REM state to the wake state is used to represent NE signal in REM state.

## Statistical analyses

Significance levels indicated are as follows: [*]$p<0.05$, [**]$p<0.01$, [***]$p<0.001$. All data are presented as mean ± SEM. All statistical tests used were two-tailed. Statistical significance was determined using both parametric (ANOVA) and non-parametric (Friedman) tests with post-hoc. All statistical testing was performed using GraphPad Prism 6.0 (GraphPad Software). No statistical methods were used to predetermine sample sizes, but our sample sizes were similar to those reported in previous publications (*Davalos et al., 2005*; *Nimmerjahn et al., 2005*).

## Acknowledgements

We thank Dr. Wei Xie from Southeast University and Dr. Hailan Hu from Zhejiang University for their careful reading and valuable comments; Li Wang and Tong Zhu from Raygenitm Biotech Co., Ltd, and Ying Guo from the company Transcend Vivoscope for comments on the optics, biological experiments, and data processing; and the Nanjing Brain Observatory for data processing services. The work was supported by grants from the National Science and Technology Innovation 2030 Major Program (2021ZD0202200, 2021ZD0202205, 2022ZD0211900, and 2022ZD0211903), the National Natural

Science Foundation of China (32293210, 92157105, 31971158, 81827809, 81827805, 82130060, and 61821002), CAMS Innovation Fund for Medical Sciences (2019-I2M-5-054), National Key Research and Development Program (2018YFA0704100 and 2018YFA0704104), Jiangsu Provincial Medical Innovation Center (CXZX202219), Collaborative Innovation Center of Radiation Medicine of Jiangsu Higher Education Institutions, Nanjing Life Health Science and Technology Project (202205045), and Key Core Technology Research Project for Nanjing Enterprise Academician Workstation.

## Additional information

### Funding

| Funder | Grant reference number | Author |
| --- | --- | --- |
| National Science and Technology Innovation 2030 Major Program | 2021ZD0202200 | Heping Cheng |
| National Science and Technology Innovation 2030 Major Program | 2021ZD0202205 | Heping Cheng Xiaochun Gu Ting Zhao |
| National Science and Technology Innovation 2030 Major Program | 2022ZD0211900 | Xianhua Wang |
| National Science and Technology Innovation 2030 Major Program | 2022ZD0211903 | Xianhua Wang Huaqiang Fang |
| National Natural Science Foundation of China | 32293210 | Heping Cheng |
| National Natural Science Foundation of China | 92157105 | Xianhua Wang |
| National Natural Science Foundation of China | 31971158 | Xianhua Wang |
| National Natural Science Foundation of China | 81827809 | Xianhua Wang |
| National Natural Science Foundation of China | 81827805 | Xiaochun Gu |
| National Natural Science Foundation of China | 82130060 | Xiaochun Gu |
| National Natural Science Foundation of China | 61821002 | Xiaochun Gu |
| Chinese Academy of Medical Sciences | 2019-I2M-5-054 | Heping Cheng Xiaochun Gu Lifeng Zhang Huaqiang Fang Xianhua Wang |
| National Key Research and Development Program | 2018YFA0704100 | Xiaochun Gu |
| National Key Research and Development Program | 2018YFA0704104 | Xiaochun Gu |
| Jiangsu Provincial Medical Center | CXZX202219 | Xiaochun Gu |
| Collaborative Innovation Center of Radiation Medicine of Jiangsu Higher Education Institutions | | Xiaochun Gu |

| Funder | Grant reference number | Author |
| --- | --- | --- |
| Nanjing Life Health Science and Technology Project | 202205045 | Xiaochun Gu |
| Key Core Technology Research Project for Nanjing Enterprise Academician Workstation | | Heping Cheng Xiaochun Gu Lifeng Zhang Huaqiang Fang Ting Zhao |

The funders had no role in study design, data collection and interpretation, or the decision to submit the work for publication.

## Author contributions

Xiaochun Gu, Conceptualization, Resources, Data curation, Formal analysis, Supervision, Funding acquisition, Validation, Investigation, Visualization, Methodology, Writing – original draft, Project administration, Writing – review and editing; Zhong Zhao, Data curation, Formal analysis, Validation, Visualization, Writing – review and editing; Xueli Chen, Data curation, Writing – review and editing; Lifeng Zhang, Huaqiang Fang, Resources, Writing – review and editing; Ting Zhao, Resources, Project administration, Writing – review and editing; Shenghong Ju, Writing – review and editing; Weizheng Gao, Xiaoyu Qian, Formal analysis, Writing – review and editing; Xianhua Wang, Funding acquisition, Writing – review and editing; Jue Zhang, Supervision, Writing – review and editing; Heping Cheng, Conceptualization, Resources, Supervision, Funding acquisition, Methodology, Writing – original draft, Project administration, Writing – review and editing

## Author ORCIDs

Xiaochun Gu ⓘ http://orcid.org/0000-0002-7289-8054
Zhong Zhao ⓘ http://orcid.org/0000-0003-3775-3818
Ting Zhao ⓘ http://orcid.org/0000-0002-9861-5999
Xianhua Wang ⓘ http://orcid.org/0000-0002-2016-9415
Heping Cheng ⓘ http://orcid.org/0000-0002-9604-6702

## Ethics

All experimental protocols were carried out with the approval of the Institutional Animal Care and Use Committee of PKU-Nanjing Institute of Translational Medicine (Approval ID: IACUC-2021-023).

Reviewer #1 (Public Review): https://doi.org/10.7554/eLife.86749.3.sa1
Reviewer #2 (Public Review): https://doi.org/10.7554/eLife.86749.3.sa2
Author Response https://doi.org/10.7554/eLife.86749.3.sa3

# Additional files

## Supplementary files
• MDAR checklist

## Data availability

Data related to the manuscript are available according to the FAIR principles via Dryad.

The following dataset was generated:

| Author(s) | Year | Dataset title | Dataset URL | Database and Identifier |
| --- | --- | --- | --- | --- |
| Gu X, Cheng H | 2023 | Data from: Imaging Microglia Surveillance during Sleep-wake Cycles in Freely Behaving Mice | https://doi.org/10.5061/dryad.f4qrfj72s | Dryad Digital Repository, 10.5061/dryad.f4qrfj72s |

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
