## [Editor Report · eLife assessment]

This **important** study uses cutting-edge miniature two-photon microscopy to follow the structural dynamics of microglia in the somatosensory cortex of freely-moving mice across the sleep/wake cycle. **Solid** evidence revealed the brain-state-dependent regulation of microglial activity, highlighting alterations in microglial morphology during REM and NREM sleep phases compared to wakefulness. Furthermore, this study provides **important** evidence for a critical role of norepinephrine from the locus coeruleus as a modulator of microglial morphology through the β2-adrenergic receptor (b2AR). Overall, the article is an **exceptional** technical feat to bridge a crucial gap in understanding sleep state-induced dynamics of microglia and its modulation by norepinephrine signaling.

---

## [Referee Report · Reviewer #1 (Public Review)]

Microglia are increasingly recognized as playing an important role in shaping the synaptic circuit and regulating neural dynamics in response to changes in their surrounding environment and in brain states. While numerous studies have suggested that microglia contribute to sleep regulation and are modulated by sleep, there has been little direct evidence that the morphological dynamics of microglia are modulated by the sleep/wake cycle. In this work, Gu et al. applied a recently developed miniature two-photon microscope in conjunction with EEG and EMG recording to monitor microglia surveillance in freely-moving mice over extended period of time. They found that microglia surveillance depends on the brain state in the sleep/wake cycle (wake, non-REM, or REM sleep). Furthermore, they subjected the mouse to acute sleep deprivation, and found that microglia gradually assume an active state in response. Finally, they showed that the state-dependent morphological changes depend on norepinephrine (NE), as chemically ablating noradrenergic inputs from locus coeruleus abolished such changes; this is in agreement with previous publications. The authors also showed that the effect of NE is partially mediated by β2-adrenergic receptors, as shown with β2-adrenergic receptor knock-out mice. Overall, this study is a technical tour de force, and its data add valuable direct evidence to the ongoing investigations of microglial morphological dynamics and its relationship with sleep. Nevertheless, microglial morphodynamics likely reflect the integrated influence of neighboring neuronal activities and neuromodulatory factors; the pan-tissue β2AR knockout mouse model may also broadly affect the animal's physiology and sleep behavior. Therefore, future studies are needed to address the specific role of microglial β2AR on its morphodynamics in sleep.

---

## [Referee Report · Reviewer #2 (Public Review)]

The MS describes an approach to monitor microglial structural dynamics and correlate it to ongoing changes in brain state during sleep-wake cycles. The main novelty here is the use of miniaturized 2p microscopy, which allows tracking microglia surveillance over long periods of hours, while the mice are allowed to freely behave. Accordingly, this experimental setup would permit to explore long-lasting changes in microglia in more naturalistic environment, which were previously not possible to identify otherwise. The findings provide key advances to the research of microglia during natural sleep and wakefulness, as opposed to anesthesia. The main findings of the paper are that microglia increase their process motility and surveillance during REM and NREM sleep as compared to the awake state. The authors further show that sleep deprivation induces opposite changes in microglia dynamics- limiting their surveillance and size. The authors then demonstrate potential causal role for norepinephrine secretion from the locus coeruleus (LC) which is driven by beta 2 adrenergic receptors (b2AR) on microglia. '

The authors have nicely demonstrated and technically validated their main conclusions. In particular, they demonstrate the utility of miniaturized 2p imaging for long lasting imaging of microglia structural changes according to sleep state over the time course of hours. The authors have done a good job in addressing all my previous concerns and provide sound evidence for sleep state induced dynamics of microglia, which is modulated by NE and depends on b2AR.

One impressive point is the ability to longitudinally track the same microglial cells in the field of view for many hours, which is highly valuable and was impossible to achieve with head fixed imaging.

The authors support their observation by using a global b2AR KO mice, which ravel impaired microglial dynamics during sleep states.

While previous evidence supports high expression and function of b2AR in microglia, these receptors are expressed throughout the brain and periphery. Therefore, the authors correctly state that the current data they show, using global b2AR KO mice, cannot be used to state a direct effect on microglia dynamics and this would warrant future experiments with cell-specific genomic manipulation.

To summarize, the main conclusions of the paper are well validated and supported with the experimental layout and analysis.

---

## [Author Response]

The following is the authors’ response to the original reviews.

Response to the Referee CommentsWe would like to express our appreciation to the editor and the reviewers for their thoughtful comments and constructive suggestions on the manuscript. We agree with most of the comments and have carefully revised the manuscript accordingly. The revisions are highlighted in red font in the revised manuscript. Below are point-by-point responses to the referee’s comments.

**Public Reviews:**

**Reviewer #1 (Public Review):**
Microglia are increasingly recognized as playing an important role in shaping the synaptic circuit and regulating neural dynamics in response to changes in their surrounding environment and in brain states. While numerous studies have suggested that microglia contribute to sleep regulation and are modulated by sleep, there has been little direct evidence that the morphological dynamics of microglia are modulated by the sleep/wake cycle. In this work, Gu et al. applied a recently developed miniature two-photon microscope in conjunction with EEG and EMG recording to monitor microglia surveillance in freely-moving mice over extended period of time. They found that microglia surveillance depends on the brain state in the sleep/wake cycle (wake, non-REM, or REM sleep). Furthermore, they subjected the mouse to acute sleep deprivation, and found that microglia gradually assume an active state in response. Finally, they showed that the state-dependent morphological changes depend on norepinephrine (NE), as chemically ablating noradrenergic inputs from locus coeruleus abolished such changes; this is in agreement with previous publications. The authors also showed that the effect of NE is partially mediated by β2-adrenergic receptors, as shown with β2-adrenergic receptor knock-out mice. Overall, this study is a technical tour de force, and its data add valuable direct evidence to the ongoing investigations of microglial morphological dynamics and its relationship with sleep. However, there are a number of details that need to be clarified, and some conclusions need to be corroborated by more control experiments or more rigorous statistical analysis. Specifically:1. The number of branch points per microglia shown here (e.g., Fig. 2g) is much lower than the values of branch points in the literature, e.g., Liu T et al., Neurobiol. Stress 15: 100342, 2021 (mouse dmPFC, IHC); Liu YU et al., Nat. Neurosci. 22: 1771-81, 2019 (mouse S1, in vivo 2P imaging). The authors need to discuss the possible source of such discrepancy.

Thank you for raising this important point. Two reasons may account for this difference. Firstly, the difference in the definition of branch points in the software. Liu YU et al. used the Sholl analysis of image J software to analyze the number of branch points of microglia. Sholl analysis defines the number of branch points as the number of crossings between branches and concentric circles of increasing radii. We reconstructed microglia morphology using Imaris, a software that defines branching points based on the number of bifurcation points. The number of bifurcations calculated represents the number of microglia branch points. Secondly, this and previous studies found that more branching points present in the state of anesthesia. The morphological characteristics of microglia in head-fixed mice under anesthesia was reported by Liu T et al. and the microglia reconstruction results presented by the authors are indeed more complex than ours.In short, this is an aspect that we have been paying attention to, and the main reasons for this difference may lie in the definition of branch points, analysis methods and related choice of thresholds. True differences in brain states and the heterogeneity of microglia in different brain regions may also contribute to the apparent discrepancy.

1. Microglia process end-point speed (Fig. 2h, o): here the authors show that the speed is highest in the wake state and lowest in NREM, which agrees with the measurement on microglia motility during wakefulness vs NREM in a recent publication (Hristovska I et al., Nat. Commun. 13: 6273, 2022). However, Hristovska et al. also reported lower microglia complexity in NREM vs wake state, which seems to be the opposite of the finding in this paper. The authors need to discuss the possible source of such differences.

This is also an important point. Hristovska et al. reported the morphodynamic characteristics of microglia during wakefulness and NREM sleep. It is worth noting that the sleep state of the mice in their experiments was unnatural due to the head fixation and body limitations, the duration of NREM sleep (sleep stability) being quite different from the NREM sleep analyzed under natural sleep. The limitations of this approach are also discussed by Hristovska et al. “Even though sleep episodes were, as anticipated, shorter than those observed in freely moving animals, changes in neuronal activity characteristic of NREM sleep were monitored by EEG recordings, and changes in morphodynamics were observed during single episodes. Several episodes of REM sleep were detected, but they were too short and rare to be analyzed reliably.” The unnatural sleep state would lead to an increase in the microarousal state, and ultimately a change in the structure of the sleep state, which may be the main reason for the difference in microglia behavior from our natural sleep.We have discussed this in the revised manuscript. Please see line 292298.

1. Fig. 3: the authors used single-plane images to analyze the morphological changes over 3 or 6 hours of SD, which raises the concern that the processes imaged at the baseline may drift out of focus, leading to the dramatic reduction in process lengths, surveillance area, and number of branch points. In fact, a previous study (Bellesi M et al., J. Neurosci. 37(21): 5263-73, 2017) shows that after 8 h SD, the number of microglia process endpoints per cell and the summed process length per cell do not change significantly (although there is a trend to decline). The authors may confirm their findings by either 3D imaging in vivo, or 3D imaging in fixed tissue.

Three lines of evidence indicate that microglia morphology changes in Fig 3 are due to SD, rather than variations in the focal plane. First, our single-plane images were quite stable over 3 or 6 hours of SD, though occasional reversible drifts might happen due to sudden motions. Second, per your suggestion, further experiments and analysis of 3D imaging were performed to monitor microglia dynamics during sleep deprivation. The new result is shown in revised Fig. S3 C-D: the length of microglia branches and the number of branching points were significantly reduced after SD, in agreement with the results of single-plane imaging. Furthermore, we detected no significant difference in microglia branching characteristics during 6h sleep deprivation in β_2_AR KO mice (Fig.S4), and this indirectly affirmed that singleplane imaging is stable enough for detecting true changes in branching during SD.

1. Fig. 4b: the EEG and EMG signals look significantly different from the example given in Fig. 2a. In particular, the EMG signal appears completely flat except for the first segment of wake state; the EEG power spectrum for REM appears dark; and the wake state corresponds to stronger low frequency components (below ~ 4 Hz) compared to NREM, which is the opposite of Fig. 2a. This raises the concern whether the classification of sleep stage is correct here.

Thank you for insightful comments. We carefully examined the behavioral video of Figure 4b, there were occasionally microarousal events indicated by slow head rotation during NREM sleep, while the companion EMG signals were completely flat, which is atypical during sleep wake cycle. The microarousal events were not excluded from sleep, which makes this set of data unrepresentative and contrary to Fig.4b. In our revised manuscript, we replaced it with more representative data that can clearly and consistently distinguish between different brain states in mice on EMG and EEG. Please see revised Fig.2a, page 34; revised Fig.4b, page 37.

1. Fig. 4 NE dynamics.• How long is a single continuous imaging session for NE?• When monitoring microglia surveillance, the authors were able to identify wake or NREM states longer than 15 min, and REM states longer than 5 min. Here the authors selected wake/NREM states longer than 1 min and REM states longer than 30 s. What makes such a big difference in the time duration selected for analysis?• Also, the definition of F0 is a bit unclear. Is the same F0 used throughout the entire imaging session, or is it defined with a moving window?

A single continuous session of NE imaging usually took about 1 hour. Subsequent analysis was performed on imaging data from each recording that included wake, NREM sleep, and REM sleep. Because of the different time scales of microglia morphological dynamic (relatively slow) and NE signals (fast), we used different time windows in the previous analysis in the previous version of the manuscript.

Per your suggestion, we have now set the same time window selection criteria for both microglia morphological and NE dynamic analysis: for wake and NREM sleep durations longer than 1 minute, and REM sleep durations longer than 30 seconds. We updated the Methods and all statistics in related figures, please see line 151-154, 481-485, 490-492; Fig. 2e-g and 2l-n, page 34. F0 definition is now explained in the Methods section. Please see line 521-522.

1. Fig. 5b: how does the microglia morphology in LC axon ablation mice compare with wild type mice under the wake state? The text mentioned "more contracted" morphology but didn't give any quantification. Also, the morphology of microglia in the wake state (Fig. 5b) appears very different from that shown in Fig. S3C1 (baseline). What is the reason?

The morphology of microglia is indeed heterogeneous and variable, affected by factors including brain state, brain region, microenvironmental changes, along with animal-to-animal difference. We didn’t perform the microglia morphology comparison between the LC axon ablation mice and wild type mice and, in view of this, we removed the description of “more contracted morphology” from the main text. It should also be noted that, as we primarily focused on changes of a microglia in different states over time by selfcomparison, we minimized possible effects of heterogeneity in microglia morphology on our conclusions.

1. The relationship between NE level and microglia dynamics. Fig. 4C shows that the extracellular NE level is the highest in the wake state and the lowest in REM. Previous studies (Liu YU et al., Nat. Neurosci. 22(11):1771-1781, 2019; Stowell RD et al., Nat. Neurosci. 22(11): 1782-1792, 2019) suggest that high NE tone corresponds to reduced microglia complexity and surveillance. Hence, it would be expected that microglia process length, branch point number, and area/volume are higher in REM than in NREM. However, Fig. 2l-n show the opposite. How should we understand this ?

Your point is well-taken. On the one hand, our data clearly showed that NE is critically involved in the brain state-dependent microglia dynamic surveillance, with evidence from the ablation of the LC-NE projection and from the β2AR knockout animal model.

On the other hand, we also understand that NE is not the sole determinant, so the relationship between the NE level and the complexity and surveillance may not be unique.

In this regard, other potential modulators also present dynamic during sleepwake cycle and may partake in the regulation of microglia dynamic surveillance. previous studies (Liu YU et al., 2019; Stowell RD et al., 2019) have shown that microglia can be jointly affected by surrounding neuronal activity and NE level during wake. It has been reported that LC firing stops (Aston-Jones et al., 1981; Rasmussen et al., 1986), while inhibitory neurons, such as PV neurons and VIP neurons, become relatively active during REM sleep (Brécier et al., 2022). ATP level in basal forebrain is shown to be higher in REM than NREM (Peng et al., 2023). In addition, our own preliminary result (Author response image 1) also showed a higher adenosine level in REM than NREM in somatosensory cortex. Last but not the least, we found that β2AR knockout failed to abolish microglial responses to sleep state switch and SD stress altogether.

In brief, microglia are highly sensitive to varied changes in the surrounding environment, and many a modulator may participate in the microglia dynamic during sleep state. This may underlie the microglia complexity difference between REM and NREM. Future investigations are warranted to delineate the signal-integrative role of microglia in physiology and under stress. We have discussed the pertinent points in the revised manuscript. Please see line 343-354.

**Author response image 1. sa3fig1:** Extracellular adenosine levels in somatosensory cortex in different brain states. AAV2/9-hSyn-GRABAdo1.0 (Peng W. et al., Science. 2020) was injected into the somatosensory cortex (A/P, -1 mm; M/L, +2 mm; D/V, -0.3 mm). Data from the same recording are connected by lines. n = 9 from 3 mice.

**Reviewer #2 (Public Review):**
The manuscript describes an approach to monitor microglial structural dynamics and correlate it to ongoing changes in brain state during sleep-wake cycles. The main novelty here is the use of miniaturized 2p microscopy, which allows tracking microglia surveillance over long periods of hours, while the mice are allowed to freely behave. Accordingly, this experimental setup would permit to explore long-lasting changes in microglia in a more naturalistic environment, which were previously not possible to identify otherwise. The findings could provide key advances to the research of microglia during natural sleep and wakefulness, as opposed to anesthesia. The main findings of the paper are that microglia increase their process motility and surveillance during REM and NREM sleep as compared to the awake state. The authors further show that sleep deprivation induces opposite changes in microglia dynamics- limiting their surveillance and size. The authors then demonstrate potential causal role for norepinephrine secretion from the locus coeruleus (LC) which is driven by beta 2 adrenergic receptors (b2AR) on microglia. However, there are several methodological and experimental concerns which should be addressed.The major comments are summarized below:1. The main technological advantage of the 2p miniaturized microscope is the ability to track single cells over sleep cycles. A main question that is unclear from the analysis and the way the data is presented is: are the structural changes in microglia reversible? Meaning, could the authors provide evidence that the same cell can dynamically change in sleep state and then return to similar size in wakefulness? The same question arises again with the data which is presented for anesthesia, is this change reversible?

As revealed by long-term free behavioral mTPM imaging, the brain-statedependent morphological changes in microglia were reproducible and reversible. Author response image 2 shows that microglia displayed reversible dynamic changes during multiple rounds of sleep-wake transition. Author response image 3 shows that microglia dynamics induced by anesthesia also exhibited reversibility.

**Author response image 2. sa3fig2:** Long-term tracking of microglia process area in different brain states. Data analysis used 8 cells. Data total of 31 time points were selected from in vivo imaging data and were used to characterize the morphological changes of microglia over a continuous 7-hour period.

**Author response image 3. sa3fig3:** Reversible changes of microglial process length, area, number of branch points under anesthesia. Wake group: 30 minute-accommodation to new environment; Isoflurane group: 1.5% in air applied at a flow rate of 0.4 L/min for 30 minutes; Recovery group: 30 minutes after recovery from anesthesia. n = 9 cells from 3 mice for each group.

1. The binary comparison between brain states is misleading, shouldn't the changes in structural dynamics compared to the baseline of the state onset? The authors method describes analysis of the last 5 minutes in each sleep/wake state. However, these transitions are directional- for instance, REM usually follows NREM, so the description of a decrease in length during REM sleep could be inaccurate.

As you know, the time scale of microglia morphological dynamic is relatively slow, so we analyzed the microglia morphological dynamic of the last part (30s in the revised manuscript) of each state instead of the state onset, allowing time for stabilization of the microglia response to inter-state transition.

Further, we compared microglia dynamic between two NREM groups transiting to different subsequent states: group1 (NREM to REM) vs group2 (NREM to Wake). This precaution was to exclude the directional effect of state transitions. Our results showed that there was no difference in microglial length, area, number of branching points between the two NREM groups (Author response image 4), indicating that the last 30s of each NREM was not affected by its following state and that it’s reasonable to perform binary comparison.

**Author response image 4. sa3fig4:** Microglial morphological length, area change, and number of branch points of the last 30s of NREM sleep followed by REM or Wake. n = 9 cells from 3 mice for each group.

1. Sleep deprivation- again, it is unclear whether these structural changes are reversible. This point is straightforward to address using this methodology by measuring sleep following SD. In addition, the authors chose a method to induce sleep deprivation that is rather harsh. It is unclear if the effect shown is the result of stress or perhaps an excess of motor activity.

We adopted the method of forced exercise as it has been commonly used for sleep deprivation (Pandi-Perumal et al., 2007; Nollet M et al., 2020), though it does have the potential limitation of excess of motor activity.

In light of your comments and suggestion, we presented new data demonstrating that sleep duration of the mice, mostly NREM sleep, increased compensatively (ZT9-10) after the 6-hour sleep deprivation (ZT2-8) (revised Fig. S3B). This result shows that sleep deprivation indeed increase sleep pressure in the mice. As the sleep pressure was eased during recovery sleep, morphological changes of microglia were reversed over a timescale of several hours (revised Fig. S3 E-J).

1. The authors perform measurements of norepinephrine with a recently developed GRAB sensor. These experiments are performed to causally link microglia surveillance during sleep to norepinephrine secretion. They perform 2p imaging and collect data points which are single neurons, and it is unclear why the normalization and analysis is performed for bulk fluorescence similar to data obtained with photometry.

We did not perform single-neuron analysis for two reasons. First, our experimental conditions, e.g., the expression of the NE indicator and the control of imaging laser intensity, did not yield sufficient signal-to-noise to clearly discriminate individual neurons with two-photon imaging. Second, NE signal may play a modulatory role, and fluorescence changes appeared to be global, rather than local or cell-specific. Therefore, we analyzed fluorescence changes in different brain states over the whole field-of-view in Fig. 4, rather than at the subregional or single-cell level.

1. The experiments involving b2AR KO mice are difficult to interpret and do not provide substantial mechanistic insight. Since b2AR are expressed throughout numerous cell types in the brain and in the periphery, it is entirely not clear whether the effects on microglia dynamics are direct. The conclusion and the statement regarding the expression of b2AR in microglia is not supported by the references the authors present, which simply demonstrate the existence and function of b2AR in microglia. In addition, these mice show significant changes in sleep pattern and increased REM sleep. This could account for reasons for the changes in microglia structure rather than the interpretation that these are direct effects.To summarize, the main conclusions of the paper require further support with analysis of existing data and experimental validation.

Previous studies have revealed that norepinephrine (NE) has a modulating effect on microglial dynamics through β2AR pathway (Stowell RD et al., 2019; Liu YU et al., 2019). Stowell et al. and Liu et al. use in vivo two-photon imaging to demonstrate that microglia dynamics differ between awake and anesthetized mice and to highlight the roles of NE and β2AR in these states (Gyoneva S et al., 2013; Stowell RD et al., 2019; Liu YU et al., 2019). To evaluate the direct effect of β2AR on microglial dynamics, Stowell et al. administered the β2AR agonist clenbuterol to anesthetized mice and found that this decreased the motility, arbor complexity, and process coverage of microglia in the parenchyma (Stowell RD et al., 2019). Inhibition of β2AR by antagonist ICI-118,551 in awake mice recapitulated the effects of anesthesia by enhancing microglial arborization and surveillance (Stowell RD et al., 2019). In addition, it has been shown microglia expressed higher numbers of β2ARs than any other cells in the brain (Zhang et al., 2014).

To this end, our current work provided new evidence to support the involvement of the LC-NE-β2AR axis in modulating microglia dynamics both during natural sleep-wake cycle and under SD stress. While we were aware the limitation of using pan-tissue β2AR knockout model that precluded us from pinpointing role of microglial β2AR, it is safe to state that β2-adrenergic receptor signaling plays a significant role in the sleep-state dependent microglia dynamic surveillance, based on the present and previous data.

We have discussed this in the revised manuscript. Please see line 324-354. As you suggested, we added references to support the statement regarding the expression of β2AR in microglia (please see line 333).

Recommendations for the authors: please note that you control which, if any, revisions, to undertake

**Reviewer #1 (Recommendations For The Authors):**
Some technical details need to be clarified. Also, please double-check for typos.1. In vivo imaging preparation: how long is the recovery time between window/EEG implantation surgery and imaging/recording?

Imaging data were collected one month after the surgery. We have added descriptions to the methods section of the revised manuscript. Please see line 419.

1. Statistical analysis: the authors used t-test or ANOVA without first checking whether the data pass the normality test. If the data does not follow a normal distribution, nonparametric tests would be more appropriate.

Per your suggestion, we performed the test of statistical significance using parametric (ANOVA) if past the normality test, or the non-parametric (Friedman) tests for non-normal data. Please see line 533-535.

1. Fig. 1b needs a minor change. In the figure, the EMG electrodes appear to be connected to the brain as well.

We have corrected this oversight. Thank you.

1. Fig. 1c: it would be helpful to give examples of raw EEG and EMG traces for REM and NREM separately.

Raw traces are now shown as suggested. Please see Fig. 1c, page 32.

1. Fig. 1h: is each data point one microglia or one end-point?

In Fig. 1h, each data represents the average speed of all branches of one microglia, not one end-point.

1. Sleep deprivation starts at 9 am. What time corresponds to Zeitgeber Time 0 (ZT0, the beginning of the light phase)?

We now clarified that 9 am corresponds to Zeitgeber time 2. Please see line 196.

1. Line 61: the authors referred to Ramon y Cajal's original suggestion that microglia dynamics are coupled to the sleep-wake cycle. However, the cited paper only indicates that Cajal suggested a role of astrocytes in the sleep-wake cycle, not microglia. In addition, there is a typo in the line: there should be a space between "Ramon" and "y" in Cajal's name.

We have updated the statement and reference literature to point out the microglia’s involvement in the sleep-wake cycle. The typo was corrected. Please see line 64-65.

1. Fig. S3B: As each group has only 3 mice, it is unclear how t-test can yield p < 0.01 or even 0.001.

We checked the original data again and it was correct. This small p-values may be due to the small intra-group difference of control group.

1. Line 251-253, "Figure 4h-n" should be "Figure 5h-n"?

We have revised it. Please see line 265-266.

1. Fig. 5h: the receptor should be "adrenergic receptor", not "adrenal receptor".

We changed the term to “adrenergic receptor”. Please see Fig 5h.

1. Fig. 5g, n: the number of data points is apparently less than the sample size given in the figure legend. Perhaps some data points have exactly the same value so they overlap? The authors may consider plotting identical values with a slight shift so that the number of data points shown matches the actual sample size, to avoid confusion.

Yes, we have added small jitters so different data points can be seen to avoid confusion. Please see Fig. 5n.

1. There are some typos (e.g., Line 217, "he" should be "the") and some incomplete references (e.g., [13], [22], [34], [35] lack volume and page number, [15] and [39] lack publisher information). Some references have inconsistent formats (e.g., "Journal of Neuroscience" is sometimes abbreviated and sometimes not). Please correct these.

We have corrected these oversights. Please see references, page 27.

**Reviewer #2 (Recommendations For The Authors):**
Major issues:1. Re-analyze the data in a manner that allows to follow and compare the same cells over different state transitions. This is necessary to evaluate the reversibility of microglia structure. In addition, consider analysis of the change from the beginning to the end of each state.

As shown in response figure 2, microglia dynamics were reversible during multiple rounds of sleep-wake transition.

1. It would be nice to see the raw data obtained over time, at least for Figure 1, before offline correction of movement to evaluate the imaging quality and level of drift during imaging.

We agree to your good suggestion. Please see the supporting material video.

1. It would be helpful to add an analysis of the percent time spent in each state for the 10 hour recordings.

Advice has been adopted. Please see revised Fig. S4C.

1. In Figure 2 the results are from 15 cells from several animals. How much do the results vary between mice? It will be helpful to show if this varies between different mice by labeling cells from each mouse differently.

In Author response image 5, in which we have labeled the distribution of data points from seven mice, there was mixed distribution of data from different animals at each brain state, but no clear animal-to-animal difference.

**Author response image 5. sa3fig5:** Quantitative analysis of microglial length based on multi-plane microglial imaging. n = 17 cells from 7 mice for each group. In right panel, each color codes data from the same animal.

1. SD- please add some quantification for sleep and EEG to show that the manipulation really caused sleep deprivation. To address the confound of forced movement and stress, it might be helpful to add quantification of movement compared to an undisturbed wakefulness.

We have added related data (revised Fig. S3B), as suggested. Please see line 196-197.

1. The DSP4 application should be also performed with NE measurements to verify the specific of the NE signal measured as well as the DSP4 toxin.

Following your suggestion, we have added DSP4 data in revised Fig. S4B.

1. Some suggested refined experiments for the b2AR KO are: a-A conditional b2AR KO in microglia, as cited in the work. b- Local application of a b2 blocker during SD. c- Imaging of NE dynamics in the b2 animals. If NE dynamics during natural sleep cycle are perturbed, then this suggests upstream mechanisms rather than direct microglia effects as suggested by the authors.

We agree that the current study cannot pinpoint a direct effect of microglia harbored β2AR. We have discussed this limitation in the revised manuscript.

Please see line 324-354.

Minor:1. Typo on page 4 (microcopy instead of microscopy).

It was corrected. Please see line 87.

1. Typo page 11- 'and he largest changes in NE' - supposed to be 'the'.

We have corrected these mistakes. Please see line 228.

1. Fig. 4- there are several units missing in the figure in panel b: the top is Hz, but what does the color bar indicate exactly? 2 what? both for theta/delta and for NE.We have modified this figure and legend for clarity. Please see Fig. 4, page 37.1. Bottom of page 12- referring to figure 4 but talking about figure 5.

The typo was corrected. Please see line 265-266.

Reference

1. Aston-Jones G, Bloom FE. Activity of norepinephrine-containing locus coeruleus neurons in behaving rats anticipates fluctuations in the sleep-waking cycle. J Neurosci. 1, 876–886 (1981).

2. Bellesi M, de Vivo L, Chini M, Gilli F, Tononi G, Cirelli C. Sleep loss promotes astrocytic phagocytosis and microglial activation in mouse cerebral cortex. J Neurosci. 37, 5263–5273 (2017).

3. Brécier A, Borel M, Urbain N, Gentet LJ. Vigilance and behavioral state-dependent modulation of cortical neuronal activity throughout the sleep/wake cycle. J Neurosci. 42, 4852–66 (2022).

4. Dworak M, McCarley RW, Kim T, Kalinchuk AV, Basheer R. Sleep and brain energy levels: ATP changes during sleep. J Neurosci. 30, 9007-16 (2010).

5. Gyoneva S., Traynelis SF. Norepinephrine modulates the motility of resting and activated microglia via different adrenergic receptors. J Biol Chem. 288, 15291302 (2013).

6. Kjaerby C, Andersen M, Hauglund N, Untiet V, Dall C, Sigurdsson B, Ding F, Feng J, Li Y, Weikop P, Hirase H, Nedergaard M. Memory-enhancing properties of sleep depend on the oscillatory amplitude of norepinephrine. Nat Neurosci. 25, 1059–1070 (2022).

7. Liu T, Lu J, Lukasiewicz K, Pan B, Zuo Y. Stress induces microglia-associated synaptic circuit alterations in the dorsomedial prefrontal cortex. Neurobiology of Stress. 15, 100342 (2021).

8. Liu YU, Ying Y, Li Y, Eyo UB, Chen T, Zheng J, Umpierre AD, Zhu J, Bosco DB, Dong H, Wu LJ. Neuronal network activity controls microglial process surveillance in awake mice via norepinephrine signaling. Nat Neurosci. 22, 1771–1781 (2019).

9. Nollet M, Wisden W, Franks NP. Sleep deprivation and stress: a reciprocal relationship. Interface Focus. 10, 20190092 (2020).

10. Pandi-Perumal SR, Cardinali DP, Chrousos GP. 2007. Neuroimmunology of sleep.New York, NY: Springer.

11. Peng W, Liu X, Ma G, Wu Z, Wang Z, Fei X, Qin M, Wang L, Li Y, Zhang S, Xu M. Adenosine-independent regulation of the sleep-wake cycle by astrocyte activity. Cell Discov. 9, 16 (2023).

12. Peng W, Wu Z, Song K, Zhang S, Li Y, Xu M. Regulation of sleep homeostasis mediator adenosine by basal forebrain glutamatergic neurons. Science. 369, 6508 (2020).

13. Rasmussen K, Morilak DA, Jacobs BL. Single unit activity of locus coeruleus neurons in the freely moving cat: I. During naturalistic behaviors and in response to simple and complex stimuli. Brain Research. 371, 324–334 (1986).

14. Stowell RD, Sipe GO, Dawes RP, Batchelor HN, Lordy KA, Whitelaw BS, Stoessel MB, Bidlack JM, Brown E, Sur M, Majewska AK. Noradrenergic signaling in the wakeful state inhibits microglial surveillance and synaptic plasticity in the mouse visual cortex. Nat Neurosci. 22, 1782-1792 (2019).

15. Umpierre AD, Bystrom LL, Ying Y, Liu YU, Worrell G, Wu LJ. Microglial calcium signaling is attuned to neuronal activity in awake mice. Elife. 27, e56502 (2020).

16. Wang Z, Fei X, Liu X, Wang Y, Hu Y, Peng W, Wang YW, Zhang S, Xu M. REM sleep is associated with distinct global cortical dynamics and controlled by occipital cortex. Nat Commun. 13, 6896 (2022).

17. Zhang Y, Chen K, Sloan SA, Bennett ML, Scholze AR, O’Keeffe S, Phatnani HP, Guarnieri P, Caneda C, Ruderisch N, Deng S, Liddelow SA, Zhang C, Daneman R, Maniatis T, Barres BA, Wu JQ. An RNA-sequencing transcriptome and splicing database of glia, neurons, and vascular cells of the cerebral cortex. J Neurosci. 34, 11929–11947 (2014).